# Characterisation of the Aroma Profile and Dynamic Changes in the Flavour of Stinky Tofu during Storage

**DOI:** 10.3390/foods12071410

**Published:** 2023-03-26

**Authors:** Huaixiang Tian, Ling Zou, Li Li, Chen Chen, Haiyan Yu, Xinxin Ma, Juan Huang, Xinman Lou, Haibin Yuan

**Affiliations:** 1Department of Food Science and Technology, Shanghai Institute of Technology, Haiquan Road 100, Shanghai 201418, China; 2Shanghai Tramy Green Food Co., Ltd., No. 201, Xuanchun Road, Sanzao Industrial Park, Xuanqiao Town, Pudong New Area, Shanghai 201314, China

**Keywords:** stinky tofu, food flavour, volatile compounds, GC-MS

## Abstract

Stinky tofu is a traditional Chinese food with wide consumption in China. Nevertheless, the dynamic changes in the flavour of stinky tofu during storage have yet to be investigated. In this study, the flavour changes of stinky tofu over six different storage periods were comprehensively analysed through sensory, electronic nose and gas chromatography-mass spectrometry (GC-MS) analyses. The results of the sensory and electronic nose analyses confirmed the changes in the flavour of stinky tofu across different storage periods. In the GC-MS analysis, 60 volatile compounds were detected during storage, and the odour activity values indicated that 29 of these 60 compounds significantly contributed to the aroma profile. During storage, the alcohol concentration of the stinky tofu gradually decreased while the acid and ester concentrations increased. According to a partial least squares analysis, 2-phenylethyl acetate, 2-phenylethyl propanoate, p-cresol, and phenylethyl alcohol, which were detected after 10 days of storage, promoting the release of an overripe apple-like odour from the stinky tofu. Findings regarding the flavour changes and characteristics of stinky tofu during different storage periods can provide a potential reference for recognising the quality of these products.

## 1. Introduction

As a type of traditional food, stinky tofu has been a popular snack since the Wei Dynasty in 220 A.D. in China [1]. Stinky tofu is prepared from soybeans, which contain a large amount of protein that decomposes into amino acids after fermentation. Moreover, stinky tofu contains vitamin B12, which can help prevent Alzheimer’s disease [2]. Stinky tofu also has a high content of s-equol, which has been proven to promote the health of menopausal women [3,4]. At present, stinky tofu has become a household snack in China and is increasingly popular.

Stinky tofu is prepared by soaking tofu in a specially fermented brine for several hours to several days. The fermented brine leads to mild fermentation of the tofu and endows it with a unique flavour [5]. This brine is typically prepared by mixing tempeh, shiitake mushrooms, amaranth, bamboo shoots, and other edible plants with water; this mixture is naturally fermented for several months to 3 years. During this period, microorganisms in the mixture grow and produce various enzymes, such as proteases and lipases, whose reaction products will contribute to the special odour profile of the brine [6]. In general, the core microbes that contributed to brine fermentation were *Enterococcus*, *Lactobacillus*, *Lactococcus* and *Leuconostoc* [5]. The characteristic aroma of stinky tofu is generated during the process of soaking tofu in fermented brine. Considering the significant influence of fermented brine on the flavour of stinky tofu, many researchers have focused on extracting and analysing the volatile compounds in different types of stinky tofu and fermented brine, for instance, by examining the characteristic flavour compounds in stinky tofu samples [7], dynamic changes in the volatile compounds in brine during fermentation [8], differences in the volatile compounds of stinky tofu sold under different brand names [9], and the flavours of fermented brine prepared by laboratory lactic fermentation [10]. 

The shelf life of stinky tofu is typically short (approximately 5–13 days). This short lifespan is attributable to the high water and protein content of tofu and the fermented brine remaining in the packaging, which leads to the continuous fermentation of products during transportation and storage. The post-fermentation process quickly and considerably changes the flavour of stinky tofu. However, none of the existing studies have investigated the nature of these changes.

In this study, volatile compounds were extracted from stinky tofu over a 13-day storage period through headspace solid-phase microextraction (HS-SPME). Moreover, gas chromatography-mass spectrometry (GC-MS) and odour active value (OAV) analyses were performed to clarify the compositions and intensities of the volatile compounds. A quantitative descriptive analysis (QDA) and an electronic nose (e-nose) analysis were performed to examine the aroma profiles of the stinky tofu during storage. We speculate that the types and concentrations of volatile flavour substances in Stinky tofu will change regularly during storage. The findings can allow researchers to compare the dynamic changes in stinky tofu flavour during storage and provide a reference for recognizing the quality of products.

## 2. Materials and Methods

### 2.1. Materials and Reagents

Alkane standards (C6–C30) were purchased from Sigma–Aldrich (St. Louis, MO, USA), and 2-octanol (internal standard, IS) was purchased from Dr. Ehrenstorfer GmbH, Augsburg, Germany. All chemicals were of analytical reagent grade with a purity higher than 98%. Stinky tofu samples were produced according to the traditional production process by soaking tofu in brine and provided by a local tofu manufacturer in Shanghai. The general production process of stinky tofu: take the fresh tofu cut into uniform blocks, completely soaked in the brine for 10–30 min. The brine was mainly fermented with amaranth and other various spices for nearly 1 year. Then the soaked tofu was immediately packaged and stored at 4 °C for sale. The storage of samples is calculated from the date of production, and the dynamic changes in aroma were examined after 1, 3, 5, 7, 10 and 13 days of storage at 4 °C.

### 2.2. Quantitative Descriptive Analysis

The sensory descriptors were determined based on international standards (ISO 8589-2007.14) after administering a check-all-that-apply survey and training the sensory evaluators, who included 19 panellists from the Shanghai Institute of Technology (10 women and 9 men aged 22–28 years), with reference to our previous study [11]. The sensory panellists were selected by assessing the sensory discrimination abilities of 32 candidates with experience in sensory testing who participated in the training process. The candidates were not informed of the purpose of the study and were trained for 15 days (60 min/day) to describe and identify aromas. Considering the descriptors recorded in previous studies and the results of preliminary experiments, descriptors agreed upon by more than 50% of the panellists were retained [12,13]. Nine sensory descriptors (overripe apple-like, rotten egg-like, mellow, winey, beany, rotten plant-like, musty, rancid, and sweaty) were reserved in the formal test. The following odorants were used to compare the nine odour descriptors: overripe apple-like, overripe apples; rotten egg-like, 0.05% hydrogen sulfide in water; mellow, cotton balls soaked with 5 mL of alcohol in a sealed container for 24 h; winey, 5 mL of whisky in a brown glass bottle; beany, fresh beans soaked in water for 12 h; rotten plant-like, stems soaked in water for 2–3 d; musty, enoki mushroom root; rancid, 0.5% butyric acid solution in propylene glycol; and sweaty, 0.01% butyric acid solution in water.

The aroma profile was analysed in an individual sensory testing laboratory at a constant temperature of 20 °C, according to the ISO 4121-2003 standard. The intensity of each aroma attribute was rated on a scale of 0–5 (0 = not perceivable, 3 = medium perceivable, 5 = strongly perceivable), with the final score of each aroma attribute corresponding to the average for all panellists.

### 2.3. E-Nose Detection

In order to verify the flavour changes of stinky tofu in different storage periods, a HERACLES e-nose from Alpha-MOS (Toulouse, France) was used to analyse the volatile compounds in the stinky tofu. The instrument was equipped with 18 metal oxide sensors and a headspace autosampler (HS100) that could perform data processing. The GC function and e-nose olfactory fingerprint software were installed in the e-nose. Stinky tofu samples (1 g) were placed in 10-mL glass vials with Teflon rubber caps. Each vial was incubated at 40 °C for 10 min with stirring (500 rpm). The headspace (5000 μL) was carried by air (150 mL/min) and injected into the nose. The sensor resistance was measured within 100 s with an acquisition frequency of once per second. The performance characteristics of electronic nose sensors are listed in Table 1.

### 2.4. Aroma Extraction through HS-SPME

The volatile compounds in the stinky tofu were extracted through HS-SPME with a divinylbenzene/carboxen/polydimethylsiloxane (DVB/CAR/PDMS) fused silica (75 μm)-coated fibre (1-cm-long; Supelco, Inc., Bellefonte, PA, USA) [14]. Prior to extraction, the stinky tofu was evenly crushed into a paste in a mortar. Subsequently, 3 g of the crushed stinky tofu and 7 μg of 2-octanol solution (220 μg/kg) were placed in a 15-mL headspace vial that was later sealed with a Teflon cover. The extraction fibre types were optimised according to the type and quantity of volatile compounds in the preliminary experiment. The number of volatile compounds extracted by 3 different SPME fibres was 34 (PDMS/DVB), 60 (DVB/CAR/PDMS), and 28 (CAR/PDMS), respectively. The final optimised extraction conditions were as follows: the headspace vial containing the stinky tofu sample was equilibrated in a water bath at 60 °C for 15 min. Subsequently, the SPME fibre was inserted in the headspace of the vial for 50 min to extract the volatile compounds. The SPME fibre was directly introduced into the injection port of the GC-MS after extraction.

### 2.5. GC-MS Analysis

The GC-MS analysis was performed using an Agilent 7890 gas chromatograph (5973C, Agilent Technologies, Santa Clara, CA, USA) equipped with a mass spectrometer. Separations were performed using HP-Innowax analytical fused silica capillary columns (60 m × 0.25 mm × 0.25 μm) from Agilent Technologies. A splitless mode was used for the injection for 4 min at 40 °C. The operating conditions for the GC-MS were as follows: the flow rate of helium was 1 mL/min, the mass spectrum was recorded in the electron impact mode (70 eV) in a scan range of 35–350 *m*/*z* with the ion source maintained at 250 °C. The initial oven temperature for the HP-Innowax column was set at 40 °C and maintained for 4 min. Subsequently, the temperature was increased to 130 °C at a rate of 4 °C/min, maintained at this value for 5 min, increased to 200 °C at a rate of 4 °C/min, and maintained at this value for 5 min [14,15]. Kovats’ retention indices (RIs) were calculated using mixtures of n-alkanes(C6–C30) for both stationary phases [16]. All volatile compounds were quantified using 2-octanol as an internal standard [17]. Compound identification was based on mass spectra matching with the standard NIST 17 MS library and on the comparison of RI sourced from the NIST Standard Reference Database. The volatile compounds were quantified by comparison of peak areas in the ion extraction chromatogram (IEC), which was obtained by selecting target ions for each compound to that of internal standard (2-octanol). These ions corresponded to base ion (*m*/*z* 100% intensity), molecular ion (M^+^) or another characteristic ion for each molecule. Hence, some peaks that could be co-eluted in scan mode can be integrated with a value of resolution greater than 1. The minimum detection limit for this method was 3 μg/kg, relative standard deviation (RSD) < 6.0. The concentration of volatile compounds was calculated as the following formula:Wi = ((Cs × Vs)/m) × (APi/APs)(1)
where Wi is the concentration of the volatile component to be measured, μg/L; Cs is the concentration of the internal standard substance, μg/L; Vs is the volume of internal standard added, μL; m is the weight of the sample, g; APi is the peak area of the volatile component; APs is the peak area of the internal standard.

### 2.6. OAV Analysis

OAV is the ratio of the concentration of an aromatic substance to its threshold value in water [18], which indicates the minimum concentration of volatile compounds that can be smelled. A compound with an OAV greater than 1 is considered to influence the aroma profile. The magnitude of this value reflects the contribution of the compound to the aroma profile. The OAV can be calculated as the following formula.
OAV = *C*_i_/*OT*_i_(2)
where *C*_i_ is the concentration of the volatile compounds, and *OT*_i_ is the threshold of the compound in water.

### 2.7. Statistical Analysis

All data were evaluated through analysis of variance (ANOVA) and Duncan’s multiple range tests, implemented using SPSS software (v. 19.0, SPSS Inc., Chicago, IL, USA). Alpha Soft (Alpha-MOS proprietary software) was used to process the e-nose data, and chemometric methods executed automatically by the software were used to interpret the e-nose measurement results. Simca-p soft (v14.1, MKS Umetrics AB) was used to derive the partial least squares (PLS) plots. Origin 2018 software (OriginLab, Northampton, MA, USA) was used to obtain radar charts of the sensory attribute scores and histograms for the GC-MS results. Each test was performed in triplicate.

## 3. Results and Discussion

### 3.1. Sensory Analysis

QDA was performed to explore the aroma profiles of the stinky tofu during the tested storage periods and differences in the sensory characteristics. Nine descriptors, namely overripe apple-like, rotten egg-like, mellow, winey, beany, rotten plant-like, musty, rancid, and sweaty, were used to analyse the aroma characteristics of the samples. As shown in Figure 1, all aroma attributes except sweaty changed significantly across the storage periods (*p* < 0.05). In the descriptive analysis, the score for beany was significantly higher (*p* < 0.05) than the scores of the other attributes in the Day 5 samples. The scores for rotten egg-like, rotten plant-like, and overripe apple-like were significantly higher, and the scores for mellow and winey were lower in the Day 13 samples than in the other samples (*p* < 0.01). Samples stored for less than 7 days exhibited similar aroma profiles. The beany attribute decreased significantly in the samples after Day 10, and the flavour profile changed significantly on Day 13. Increases in unpleasant flavour attributes, such as overripe apple-like, musty, rotten egg-like, and rotten plant-like, deteriorated the flavour of the stinky tofu. 

### 3.2. E-Nose Analysis

To verify the difference in the flavour of stinky tofu across different storage periods, a dynamic factor analysis (DFA) was performed using an e-nose to identify correlations between the individual composition variables of the stinky tofu over time [19]. The two-dimensional DFA plot is shown in Figure 2. The sum of the first two discriminant factor values was 91.936% (DF1: 54.942% and DF2: 36.994%). The stinky tofu samples stored for less than 7 days are distributed on the positive semi-axis of DF1, whereas the samples stored for less than 5 days are concentrated in the first quadrant and are easily distinguishable from the samples stored for more than 10 days. The flavour characteristics of stinky tofu samples stored for 1 to 7 days were similar, whereas significant differences were observed after 10 days of storage. Both of the stinky tofu samples stored for 10 and 13 days could be distinct from earlier time points. Overall, the results of the e-nose analysis verified the flavour of the stinky tofu changed significantly with increasing storage time, and significant flavour differences were observed between the samples in the early and late stages of storage.

### 3.3. Volatile Compounds in the Stinky Tofu Samples

The volatile compounds in the stinky tofu were analysed by HS-SPME-GC-MS in the 13-day storage period and are listed in Table 2. A total of 60 volatile compounds were detected, including 10 acids, 17 alcohols, 11 aldehydes, 13 esters, three phenols, three ketones and three compounds of other types. Among them, 29 volatile compounds with OAVs greater than 1 were detected (Table 3), including three acids, six alcohols, 10 aldehydes, six esters, two phenols, one ketone and one furan substance. In particular, 1-octen-3-ol, (E,E)-2,4-decadienal, (E,E)-2,4-nonadienal, nonanal, and p-cresol contributed significantly to the aroma profile of the stinky tofu (OAV > 100). Among these compounds with OAV greater than 100, 1-octen-3-ol typically has a mushroom-like smell, (E,E)-2,4-decadienal typically has a chicken and fatty smell, whereas the other two aldehydes have a green and fatty smell. All of these compounds except p-cresol were considered the main sources of the beany odour.

The stinky tofu over 6 different storage periods exhibited high concentrations of acids, including acetic acid, propanoic acid, and butanoic acid. The maximum concentration and OAV corresponded to butyric acid, which has a strongly acidic and sweat-like smell and likely contributes significantly to the overall aroma profile of stinky tofu. In addition, 2-methylbutanoic acid and 3-methylbutanoic acid, which have pungent, sour, and rancid odours, had high OAVs (OAV ≥ 1). The branched-chain fatty acids in stinky tofu are typically produced by the fermentation of amino acids by anaerobic microorganisms [19]. The obtained results are similar to those obtained by Wang et al. in a study of fermented brine [20].

The alcohol compounds, which are mainly derived from the fermentation of raw materials in the brine, endowed the stinky tofu with a mellow aroma and functioned as the main components of the volatile compounds in the stinky tofu. Among these alcohols, hexanol and 1-octen-3-ol may promote the beany smell of stinky tofu through synergy or masking effects [20]. Phenylethyl alcohol, which has a sweet rose-like fragrance, is a typical volatile compound in food and is likely derived from benzenesulfonic acid through a series of reactions [21,22].

Aldehydes present in the sample are widely recognized as off-flavour substances produced during food storage [23,24]. Aldehyde compounds usually exhibit green, herbal and fatty odours [20], in which benzaldehyde is a typical volatile compound in food and has a bitter almond-like nutty aroma, and hexanal is considered to be a typical compound to enhance the beany odour of the stinky tofu [25].

Esters, which typically exhibit floral and fruity scents, contributed to the ‘fermented’ and ‘over-ripe apple’ odour in the aroma profile of the stinky tofu. Although the pleasant floral and fruity aromas of esters have been noted to enhance food flavours [26], the excessive ester concentration in the Day 13 samples did not result in a pleasant odour, likely because of the dynamic trends of volatile compounds and masking of the ester aromas by acids.

Phenol and p-cresol, which exhibit unpleasant odours, were found to contribute to the characteristic odour of stinky tofu. These conclusions were consistent with those of existing studies [14,26].

There are many kinds of volatile compounds in stinky tofu, which was very similar to some existing research results. However, some compounds considered the core flavour substance of stinky tofu was not detected, such as indole and skatole [9,14]. The volatile compounds in stinky tofu may relate to the processing technology, raw materials and geographic area [5]. The stinky tofu used in this research was sourced from the southeast coastal regions of China, and the brine was fermented using an amaranth-based mixture of plants. This mixture may impart a different aroma compound than that found in the central and western regions of China.

### 3.4. Changes in the Aroma Profile during Storage

The dynamic changes in the contents and ratios of volatile compounds in the stinky tofu during 6 different storage periods were determined by comparing and analysing the volatile compounds. As shown in Figure 3, as the storage period increased, the number of volatile compounds in the stinky tofu first increased and then decreased. Furthermore, the compositions of volatile compounds in the stinky tofu stored for less than 7 days were similar, and the composition of the compounds changed significantly after 10 days. The number of volatile compounds was maximised in the Day 5 samples (40 compounds), likely because of the incomplete fermentation of the freshly prepared stinky tofu and, consequently, the incomplete production of the volatile compounds. Alcohols, esters, and phenols exhibited significant fluctuations in numbers and concentrations (*p* ≤ 0.05). The numbers and concentrations of esters increased significantly after Day 10, whereas those of alcohols decreased. In terms of concentration, the significant increases in ester concentrations in post-fermentation samples likely occurred because esters are generated by bacterial esterification, which may have been promoted by the high-water activity of the stinky tofu during storage [27]. In general, stinky tofu samples stored for different time periods had distinct flavours and a regular dynamic trend in these flavours was observed with changes in the storage time. During storage, the stinky tofu exhibited high concentrations of acids, alcohols, and aldehydes. The acid concentration in nearly all samples was high and contributed significantly to the aroma profile of the stinky tofu owing to their low threshold. The acid concentrations peaked (72.8%) in the Day 3 sample. The ester concentrations increased significantly after Day 7 (*p* < 0.05), and extremely high concentrations were observed in the Day 13 sample. These high concentrations likely occurred because the acid produced in the later stage of fermentation provided a sufficient substrate for the formation of esters [28].

As shown in Figure 3, esters were mainly detected in the post-fermentation samples (storage period of 10–13 days), and the numbers of esters increased as the storage period was prolonged, likely because of the delay in esterification between acids and alcohols [22]. High concentrations of the esters 2-phenethyl acetate and 2-phenethyl propionate were detected on Day 7 and Day 10. Notably, these two volatile compounds, which represent the final products of benzenesulfonic acid conversion, only appeared after Day 10 and may indicate over-fermentation of the stored stinky tofu [21]. The p-cresol was observed in all storage periods, and its concentrations increased with the storage time, with the maximum values observed in the Day 13 sample. The p-cresol exhibits a phenolic odor, and it has been reported that it can contribute to the unique aroma of stinky tofu [13].

### 3.5. PLS Analysis

PLS analyses were performed to examine the correlations between the sensory attributes and the volatile compounds of the stinky tofu. The concentrations of 29 volatile compounds with OAV greater than 1 were represented as the X-matrix, and the nine sensory descriptors were represented as the Y-matrix. Among these 29 compounds, 15 compounds exhibited VIP values greater than 1, indicating the significant contribution of these compounds to the aroma of the stinky tofu. Volatile compounds and sensory analysis data were used to establish a model to identify the relationships between the volatile compounds and sensory attributes (Figure 4). The results indicated that most of the sensory attributes were correlated with the volatile compounds, and nearly all X and Y variables lay within the ellipse (R^2^ = 100%; R^2^ represents the degree of interpretation).

The concentrations of volatile compounds that contributed to beany flavour were higher in the early periods of storage, such as 1-hexanol (2), 2-pentyl-furan (29), hexanal (14), decanal (18), (E)-2-nonenal (19) and (E,E)-2,4-nonadienal (21), which consistent with previous results [21]. In addition, ethyl acetate (7), propanoic acid ethyl ester (8), acetic acid pentyl ester (9), acetic acid hexyl ester (10), propanoic acid hexyl ester (11) and 2-phenylethyl acetate (12), were considerably influenced the score for the overripe apple-like attribute. It is worth mentioning that all of the substances related to overripe apple-like flavor were esters, with rapidly increased concentrations in the late fermentation stage. Generally, the sweet, fruity flavour of esters had a positive effect on food flavour [29]; however, the sensory of esters was perceived as an unpleasant “overripe” odour in the samples, likely due to the inadequate coordination of esters with other volatile compounds in stinky tofu. This finding is also consistent with the results of sensory analysis for stinky tofu samples. Butyric acid (23) was significantly correlated with the descriptors of rotten eggs and rancidity in the samples, and the point representing p-cresol (28) was close to the descriptors of “Rottenplant.” These two compounds were demonstrated to be strongly associated with the characteristic odor of stinky tofu, a finding that is similar to those reported in other studies [21]. Therefore, the PLSR analysis results and the key aroma compounds identified by OAV indicated that the different types and concentrations of volatile compounds have a significant impact on the aroma attributes of stinky tofu.

## 4. Conclusions

The aroma profile and dynamic changes in the flavour of the stinky tofu during storage were investigated based on QDA and SPME-GC-MS. Sixty volatile compounds were isolated and identified in stinky tofu samples stored for six periods by SPME-GC-MS, and the contributions of these compounds to the aroma were investigated by calculating the OAVs. According to the results of sensory evaluation, the scores of unpleasant flavour attributes such as sweaty, rancid, rotten plant-like and rotten egg-like tended to increase at the late stage of storage. Based on the PLS analysis, the esters in the stinky tofu generated an overripe apple-like aroma, and stinky tofu stored for more than 7 days exhibited significant increases in the ester concentration. The ester concentration thus can be considered an indicator of the product storage time. The concentrations of 1-hexanol, 2-pentyl-furan, hexanal, decanal, (E)-2-nonenal and (E,E)-2,4-nonadienal were positively correlated with beany characteristics and were higher in the early stages of storage than in later stages. The results regarding the flavour changes and characteristics of stinky tofu during different storage periods can provide a reference for recognizing the quality of products. In addition, the established correlation between aroma compounds and sensory descriptors of stinky tofu can be leveraged for further research, such as targeted regulation of certain aroma characteristics in stinky tofu based on the concentrations of volatile compounds associated with descriptors.

## Figures and Tables

**Figure 1 foods-12-01410-f001:**
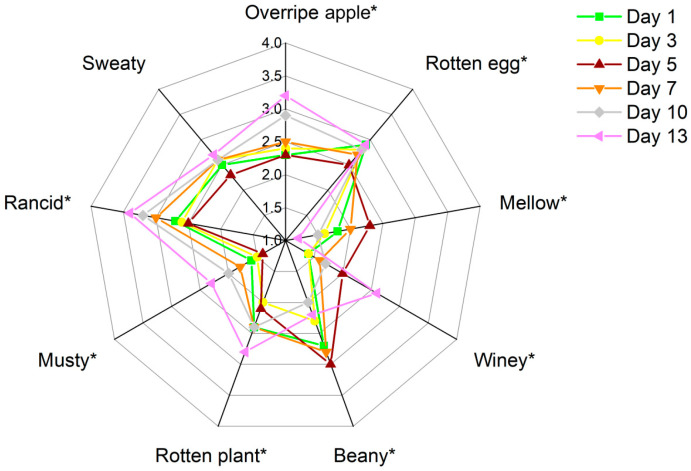
Sensory aroma profiles of the stinky tofu samples in different storage periods. ‘*’ indicates the significance (*p* < 0.05) according to ANOVA and Duncan’s multiple comparison tests.

**Figure 2 foods-12-01410-f002:**
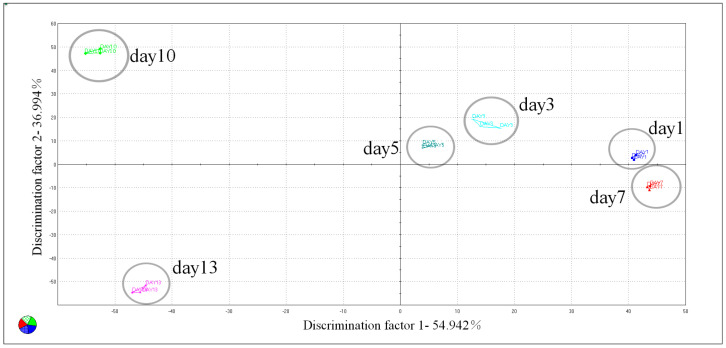
Results of e–nose of DFA plot of the stinky tofu samples in different storage periods.

**Figure 3 foods-12-01410-f003:**
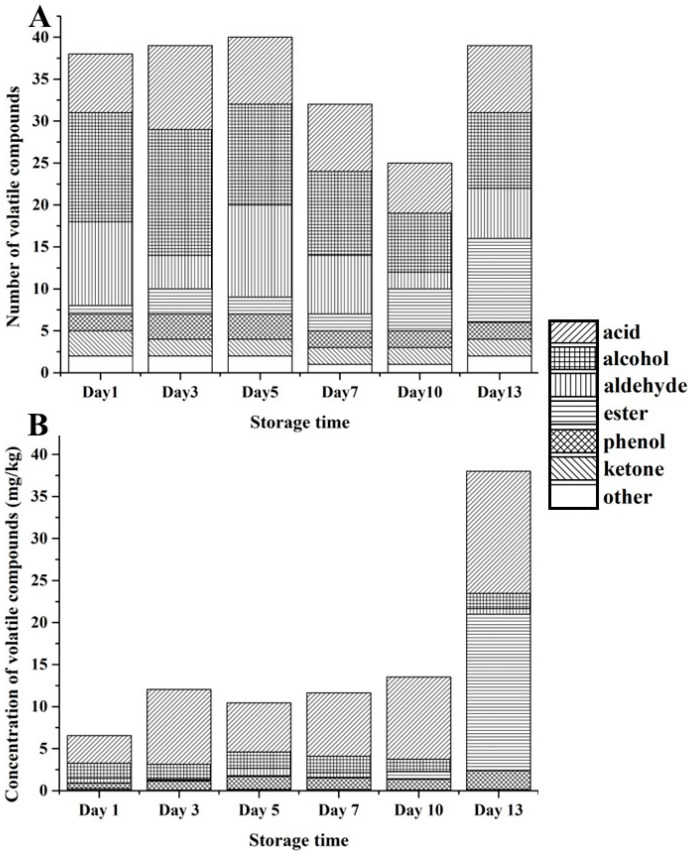
(**A**) Numbers and (**B**) concentrations of volatile compounds in the stinky tofu samples during storage.

**Figure 4 foods-12-01410-f004:**
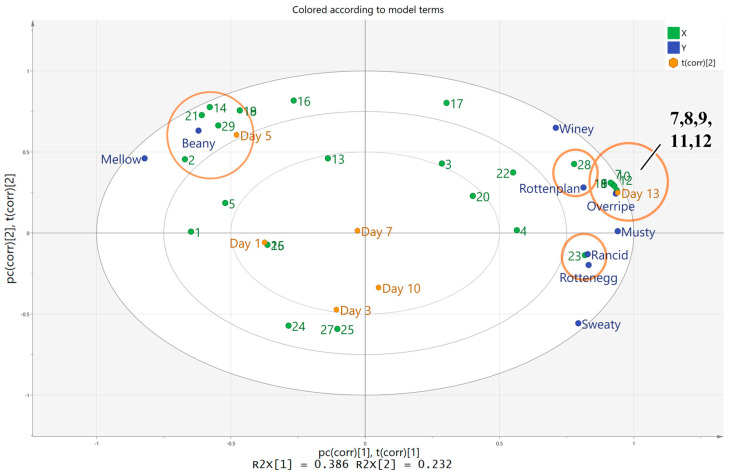
Partial least squares correlation plot from the analysis of the stinky tofu samples. The model was derived from 29 volatile compounds with OAV greater than 1 as the X–matrix and the result of sensory analysis as the Y–matrix. Codes 1–29 refer to the volatile compounds listed in Table 3.

**Table 1 foods-12-01410-t001:** Performance characteristics of electronic nose sensors.

No.	Sensor Name	Type of Sensitive Substance
1	LY2/LG	Chlorine, Fluorine, Sulfide
2	LY2/G	Ammonia, Amine compounds
3	LY2/AA	Ethanol, Ammonia
4	LY2/Gh	Ammonia, Amine compounds
5	LY2/gCTI	Sulfide
6	LY2/gCT	Propane, Butane
7	T30/1	Propanol, Hydrogen chloride
8	P10/1	Hydrocarbons, n-octane
9	P10/2	Methane, n-heptane
10	P40/1	Fluorine, Chlorine, Methyl furfural
11	T70/2	Xylene, Toluene
12	PA/2	Acetaldehyde, Amine compounds
13	P30/1	Ammonia, Ethanol
14	P40/2	Chlorine, Methyl mercaptan
15	P30/2	Hydrogen sulfide, Copper
16	T40/2	Chlorine
17	T40/1	Fluorine
18	TA/2	Alcohol

**Table 2 foods-12-01410-t002:** The volatile compounds detected by GC-MS in stinky tofu during storage.

No.	RI **		Compound	Concentration (mg/kg) *					
	Calc	Ref		Day 1	Day 3	Day 5	Day 7	Day 10	Day 13
1	1455	1461	Acetic acid	1.014 ± 0.822 ^a^	1.723 ± 0.132 ^a^	2.035 ± 0.242 ^a^	2.754 ± 0.003 ^c^	3.299 ± 0.597 ^b^	3.471 ± 0.320 ^c^
2	1541	1564	Propanoic acid	0.498 ± 0.310 ^a^	0.652 ± 0.319 ^bc^	0.906 ± 0.161 ^ab^	1.212 ± 0.010 ^c^	1.588 ± 0.467 ^c^	2.825 ± 0.258 ^d^
3	1571	1544	2-methyl-propanoic acid	0.052 ± 0.019 ^b^	0.052 ± 0.013 ^b^	0.082 ± 0.010 ^ab^	0.088 ± 0.008 ^b^	0.036 ± 0.003 ^a^	0.027 ± 0.006 ^b^
4	1630	1639	Butanoic acid	1.367 ± 0.555 ^a^	2.495 ± 1.201 ^c^	3.471 ± 0.666 ^b^	4.173 ± 0.124 ^b^	5.692 ± 0.418 ^c^	7.501 ± 0.360 ^d^
5	1672	1674	2-methyl-butanoic acid	0.182 ± 0.048 ^a^	0.374 ± 0.012 ^b^	0.263 ± 0.031 ^b^	0.118 ± 0.018 ^b^	—	—
6	1673	1679	3-methyl-butanoic acid	—	0.331 ± 0.060	—	—	—	—
7	1738	1734	Pentanoic acid	0.066 ± 0.015 ^a^	0.287 ± 0.051 ^c^	0.273 ± 0.007 ^a^	0.164 ± 0.007 ^a^	0.067 ± 0.018 ^a^	0.019 ± 0.012 ^b^
8	1801	1800	4-methyl-pentanoic acid	—	0.021 ± 0.001 ^ab^	0.099 ± 0.003 ^c^	0.167 ± 0.013 ^bc^	0.149 ± 0.020 ^c^	0.132 ± 0.012 ^abc^
9	1843	1831	Hexanoic acid	0.125 ± 0.055 ^a^	0.266 ± 0.071 ^bc^	0.17 ± 0.047 ^ab^	0.162 ± 0.006 ^ab^	0.105 ± 0.039 ^a^	0.066 ± 0.004 ^cd^
10	2055	2039	Octanoic acid	—	0.038 ± 0.014 ^ab^	0.018 ± 0.002 ^a^	—	—	—
11	928	939	Ethanol	0.686 ± 0.087 ^bc^	0.585 ± 0.090 ^ab^	0.532 ± 0.088 ^ab^	0.553 ± 0.006 ^ab^	0.541 ± 0.490 ^c^	0.309 ± 0.035 ^a^
12	1033	1037	1-propanol	—	0.011 ± 0.002	—	0.009 ± 0.001	—	—
13	1142	1150	1-butanol	—	0.014 ± 0.001	—	0.012 ± 0.001	—	—
14	1205	1206	3-methyl-1-butanol	0.041 ± 0.028 ^c^	0.029 ± 0.002 ^b^	0.011 ± 0.001 ^a^	—	—	—
15	1246	1254	1-pentanol	0.016 ± 0.000 ^a^	0.023 ± 0.014 ^b^	0.022 ± 0.017 ^b^	0.022 ± 0.003 ^c^	—	—
16	1349	1340	1-hexanol	0.704 ± 0.316 ^cd^	0.765 ± 0.013 ^bc^	0.827 ± 0.022 ^d^	0.955 ± 0.107 ^d^	0.234 ± 0.097 ^a^	0.157 ± 0.003 ^ab^
17	1389	1394	3-octanol	0.035 ± 0.004 ^ab^	0.029 ± 0.020 ^a^	0.092 ± 0.002 ^d^	0.046 ± 0.010 ^bc^	0.034 ± 0.002 ^ab^	0.011 ± 0.001 ^c^
18	1447	1447	1-octen-3-ol	0.186 ± 0.042 ^ab^	0.262 ± 0.070 ^d^	0.259 ± 0.021 ^d^	0.211 ± 0.005 ^c^	0.056 ± 0.001 ^a^	—
19	1451	1459	1-heptanol	0.021 ± 0.048 ^a^	0.022 ± 0.030 ^a^	0.034 ± 0.008 ^a^	0.051 ± 0.022 ^d^	0.223 ± 0.015 ^c^	0.044 ± 0.007 ^b^
20	1508	1504	(E)-2-hepten-1-ol	0.037 ± 0.001 ^a^	0.029 ± 0.011 ^a^	0.015 ± 0.009 ^b^	—	—	—
21	1554	1554	1-octanol	0.024 ± 0.009	0.017 ± 0.005	—	—	—	—
22	1609	1609	(E)-2-octen-1-ol	0.062 ± 0.001 ^a^	0.042 ± 0.003 ^b^	0.011 ± 0.001 ^c^	—	—	—
23	1656	1666	1-nonanol	—	0.122 ± 0.001 ^a^	0.134 ± 0.007 ^a^	0.056 ± 0.001 ^b^	0.028 ± 0.001 ^c^	—
24	1912	1935	Phenylethyl alcohol	0.018 ± 0.009 ^a^	0.044 ± 0.006 ^a^	0.042 ± 0.001 ^a^	0.064 ± 0.040 ^a^	0.093 ± 0.035 ^a^	0.304 ± 0.068 ^b^
25	1957	1935	1-dodecanol	—	—	—	—	0.128 ± 0.001	0.412 ± 0.012
26	1878	1889	Benzyl alcohol	0.032 ± 0.021 ^a^	0.021 ± 0.001 ^a^	0.025 ± 0.001 ^a^	0.027 ± 0.001 ^a^	0.031 ± 0.001 ^a^	0.049 ± 0.004 ^b^
27	1957	1935	1-Dodecanol	—	—	—	0.138 ± 0.001 ^a^	0.228 ± 0.024 ^b^	0.412 ± 0.012 ^c^
28	1528	1529	Benzaldehyde	0.208 ± 0.197 ^b^	0.165 ± 0.006 ^a^	0.128 ± 0.001 ^ab^	0.055 ± 0.438 ^a^	0.027 ± 0.001 ^a^	—
29	1712	1730	4-ethyl-benzaldehyde	0.056 ± 0.018 ^a^	0.042 ± 0.008 ^a^	0.032 ± 0.008 ^b^	0.013 ± 0.002 ^c^	—	—
30	1217	1220	(E)-2-hexenal	0.014 ± 0.000	0.002 ± 0.001	—	—	—	—
31	1322	1332	(E)-2-heptenal	0.052 ± 0.023 ^a^	0.059 ± 0.027 ^a^	0.064 ± 0.017 ^b^	0.054 ± 0.001 ^a^	—	—
32	1390	1396	Nonanal	0.049 ± 0.011 ^a^	0.062 ± 0.001 ^b^	0.189 ± 0.020 ^c^	0.049 ± 0.032 ^ab^	0.014 ± 0.001 ^b^	—
33	1429	1434	(E)-2-octenal	0.056 ± 0.000 ^a^	0.071 ± 0.001 ^a^	0.097 ± 0.007 ^b^	0.052 ± 0.001 ^a^	—	—
34	1496	1500	Decanal	—	—	0.035 ± 0.008	—	—	—
35	1535	1542	(E)-2-nonenal	—	—	0.048 ± 0.007	—	—	—
36	1642	1643	(E)-2-decenal	0.022 ± 0.001 ^a^	0.037 ± 0.008 ^a^	0.055 ± 0.438 ^b^	0.085 ± 0.001 ^c^	0.074 ± 0.011 ^c^	0.051 ± 0.001 ^b^
37	1701	1706	(E,E)-2,4-nonadienal	0.007 ± 0.001 ^a^	0.014 ± 0.002 ^a^	0.018 ± 0.002 ^b^	—	—	—
38	1809	1827	(E,E)-2,4-decadienal	0.043 ± 0.033 ^a^	0.065 ± 0.002 ^a^	0.113 ± 0.019 ^ab^	0.298 ± 0.044 ^c^	0.314 ± 0.001 ^c^	0.241 ± 0.021 ^bc^
39	889	891	Ethyl acetate	—	—	—	—	0.011 ± 0.001	0.329 ± 0.001
40	951	964	Propanoic acid ethyl ester	—	—	—	—	—	0.094 ± 0.011
41	1069	964	Acetic acid butyl ester	—	—	—	—	—	0.014 ± 0.002
42	1171	1176	Acetic acid pentyl ester	—	—	—	—	—	0.074 ± 0.002
43	1268	1269	Acetic acid hexyl ester	—	0.011 ± 0.001 ^a^	0.014 ± 0.003 ^a^	0.019 ± 0.001 ^a^	0.095 ± 0.055 ^b^	1.862 ± 0.032 ^c^
44	1336	1342	Propanoic acid, hexyl ester	—	—	—	—	—	0.327 ± 0.012
45	1349	1336	Formic acid, hexyl ester	—	—	—	—	—	0.197 ± 0.004
46	1371	1392	Acetic acid heptyl ester	—	—	—	—	—	0.018 ± 0.001
47	1554	1560	Formic acid octyl ester	—	—	0.057 ± 0.001	0.025 ± 0.003	—	—
48	1580	1579	Bornyl acetate	—	0.005 ± 0.001 ^a^	0.018 ± 0.009 ^a^	0.002 ± 0.001 ^a^	—	—
49	1638	1635	Butyrolactone	0.032 ± 0.002 ^a^	0.069 ± 0.009 ^b^	0.031 ± 0.001 ^a^	—	—	—
50	1818	1825	2-phenylethyl acetate	—	—	—	—	0.734 ± 0.169	12.172 ± 0.684
51	1884	/	2-phenyl ethyl propanoate	—	—	—	—	0.078 ± 0.022	3.515 ± 0.699
52	1864	1889	2-methoxy phenol	—	0.017 ± 0.001 ^a^	0.041 ± 0.007 ^b^	0.018 ± 0.004 ^a^	—	—
53	2009	2004	Phenol	0.119 ± 0.050 ^a^	0.125 ± 0.022 ^a^	0.178 ± 0.001 ^ab^	0.105 ± 0.514 ^ab^	0.077 ± 0.021 ^a^	0.061 ± 0.010 ^c^
54	2086	2094	P-cresol	0.367 ± 0.189 ^a^	1.014 ± 0.141 ^b^	1.316 ± 0.255 ^b^	1.716 ± 0.102 ^b^	1.807 ± 0.503 ^b^	2.031 ± 0.080 ^c^
55	1281	1278	2-octanone	0.041 ± 0.011 ^a^	0.022 ± 0.002 ^b^	0.028 ± 0.014 ^b^	0.023 ± 0.514 ^b^	0.022 ± 0.002 ^b^	0.017 ± 0.004 ^a^
56	1656	1647	Acetophenone	0.124 ± 0.012 ^a^	0.082 ± 0.008 ^b^	0.039 ± 0.009 ^b^	—	—	—
57	1968	1988	Maltol	0.019 ± 0.013 ^b^	0.026 ± 0.014 ^b^	0.022 ± 0.010 ^b^	0.045 ± 0.027 ^bc^	0.003 ± 0.002 ^a^	—
58	1224	1244	2-pentyl-furan	0.088 ± 0.032 ^cd^	0.084 ± 0.002 ^a^	0.102 ± 0.016 ^d^	0.082 ± 0.001 ^cd^	0.062 ± 0.008 ^bc^	0.046 ± 0.029 ^b^
59	1257	1257	Styrene	0.051 ± 0.016 ^a^	0.011 ± 0.008 ^b^	0.005 ± 0.001 ^c^	—	—	—
60	1618	1619	2-(2-ethoxyethoxy)-ethanol	0.046 ± 0.002 ^a^	0.077 ± 0.031 ^b^	0.031 ± 0.001 ^a^	—	—	—

Note: *: Volatile compounds identification based on the NIST17 mass spectral database. **: The retention index of volatile compounds on HP-Innowax columns. —: Not detected in the sample. /: Not found in references. Values with different letters (a to d) in a row are significantly different (*p* ≤ 0.05).

**Table 3 foods-12-01410-t003:** Thresholds and OAVs (≥1) of the volatile compounds in the stinky tofu during storage.

NO.	Compound ^A^	Threshold (mg/kg) ^B^	OAV						Description
			Day 1	Day 3	Day 5	Day 7	Day 10	Day 13	
1	3-methyl-1-butanol	0.02	2	1	1	—	—	—	Apple brandy-like, spicy
2	1-hexanol	0.07	10	7	7	6	3	2	Resin-like, flowery, green
3	1-octen-3-ol	0.0015	57	74	93	120	137	182	Mushroom-like
4	1-heptanol	0.003	7	7	—	12	—	15	Green, chemical-like
5	(E)-2-octen-1-ol	0.02	3	2	1	1	—	—	Mellow
6	1-dodecanol	0.016	—	—	—	11	16	26	Mellow
7	Ethyl acetate	0.005	—	—	—	—	21	66	Mellow, fruity
8	Propanoic acid ethyl ester	0.01	—	—	—	—	—	9	Mellow, fruity
9	Acetic acid pentyl ester	0.043	—	—	—	—	—	2	Fruity, banana- and pear-like
10	Acetic acid hexyl ester	0.002	—	6	8	10	48	931	Fruity
11	Propanoic acid, hexyl ester	0.008	—	—	—	7	21	41	Sweet, fruity
12	2-phenylethyl acetate	0.24959	—	—	—	—	13	24	Sweet, slightly green leafy
13	4-ethyl-benzaldehyde	0.013	4	2	2	—	—	2	Bitter almond-like, slightly sweet
14	Hexanal	0.005	10	17	44	26	11	—	Green, citrusy and fatty
15	(E)-2-octenal	0.003	19	9	—	—	—	—	Green, citrusy, and fatty
16	Nonanal	0.0011	45	57	172	41	17	—	Green, citrusy, and fatty
17	(E)-2-heptenal	0.013	5	5	5	4	—	7	Irritating, green grass-like
18	Decanal	0.003	—	—	10	1	—	—	Green and fatty
19	(E)-2-nonenal	0.00019	97	162	251	180	97	12	Green
20	(E)-2-decenal	0.0003	73	108	167	284	220	169	Green and fatty
21	(E,E)-2,4-nonadienal	0.0001	70	123	180	110	42	—	Green and fatty
22	(E,E)-2,4-decadienal	0.000027	1593	1928	4169	11,032	4235	2924	Spicy, geranium-like
23	Butanoic acid	0.204	7	12	17	22	28	37	Strongly sour, cheese-like
24	2-methyl-butanoic acid	0.1	2	4	3	3	—	—	Sour, cheese-like
25	3-methyl-butanoic acid	0.046	—	7	12	4	—	—	Milk-like, sour, slightly sweet
26	Acetophenone	0.065	2	—	—	—	—	—	Hawthorn-like
27	2-methoxyphenol	0.0016	3	11	18	8	—	—	Sweet, woody, slightly smoky
28	P-cresol	0.01	37	101	132	122	111	103	Phenolic, sour
29	2-pentyl-furan	0.0058	5	14	18	14	11	8	Oily, soy, green

^A^ The aroma compounds identified on the HP-Innowax column; ^B^ The threshold of volatile compounds in water referred to in the literature [19]. —: Not detected in the sample.

## Data Availability

The data presented in this study are available on request from the corresponding author.

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
