# Peer review of "Characterisation of the Aroma Profile and Dynamic Changes in the Flavour of Stinky Tofu during Storage"

_foods, 2023, doi:10.3390/foods12071410_

Round 1
Reviewer 1 Report
Authors performed interesting and complex study of stinky tofu samples during fermentation and storage. I suggest to revise, as follows:
- It was very difficult to read Figures 2 and 4 and to follow the discussion. Please, increase the quality of the figures.
- Authors wrote that the volatile compounds were quantified by comparison of peak areas in the ion extraction chromatogram. But, then, in Table 1, concentration of individual volatiles was presented as mg/kg. Please, describe how concentration was calculated. Please, write the concentration range for calibration.
- Concerning the HS-SPME and GC-MS analysis, authors should explain whether they used own methods or the experimental conditions have been taken from some publication. Please, add appropriate reference, or include some validation parameters, such as LOD, LOQ if authors used own developed methods in order to check their accuracy.
Reviewer 2 Report
The article illustrates characterisation of the aroma profile and dynamic changes in the flavour of stinky tofu during storage. There are several communication problems: from English at times difficult to understand, to the confusion about the terms and concepts used. To improve the quality of the article, the following are suggested:
First of all, the paper needs a thorough editing work since there are grammatical errors rampant throughout the paper.
Author should add the novelty of this work in the first paragraph of the introduction section and add a clear hyphothesis in the last paragraph.
This paper does not explain the contribution and motivation of this study. Why is this case critical? What's the new contribution?
Line 104: The specifications of the sensors used in the electronic nose should be added.
It is better the discussion section expand with previous research.
Authors should discuss the results and how they can be interpreted from the perspective of previous studies and of the working hypotheses. The findings and their implications should be discussed in the broadest context possible. Future research directions may also be highlighted.
Reviewer 3 Report
This paper presents in a very clear and complete way the characterisation of the aroma profile and dynamic changes in the flavour of stinky tofu during storage.
This study includes different kinds of analyses: Quantitative determination of the main volatile compounds using HS-SPME-GC-MS, sensory analyses and PLS analyses in order to find correlation between the volatile composition and the final aroma.
Suggestion
In my opinion authors could include some figure or table with data from the optimization of the different parameters of the HS-SPME extraction.
authors should be considered to include some information about quality parameters of the methodology such as: precission, calibration range, limits of detection,....
Round 2
Reviewer 2 Report
no comments